# Enzymatic Synthesis of Modified Nucleoside 5′-Monophosphates

**Martyna Koplūnaitė** *, **Kamilė Butkutė**, **Dominykas Špelveris**, **Nina Urbelienė** and **Rolandas Meškys**

Department of Molecular Microbiology and Biotechnology, Institute of Biochemistry, Life Sciences Center, Vilnius University, Sauletekio al. 7, LT-10257 Vilnius, Lithuania
* Correspondence: martyna.koplunaite@gmc.vu.lt

**Abstract:** There is an extensive list of applications for nucleosides, nucleotides, and their analogues that spans from substrates and inhibitors in enzymatic research to anticancer and antiviral drugs. Nucleoside phosphates are often obtained by chemical phosphorylation reactions, although enzymatic nucleoside phosphorylation is a promising green alternative. In this work two nucleoside kinases, *D. melanogaster* deoxynucleoside kinase and *B. subtilis* deoxycytidine kinase, have been employed for the phosphorylation of various canonical and modified nucleosides, and the results between the two enzymes have been compared. It was determined that both kinases are suitable candidates for enzymatic nucleoside 5′-monophosphate synthesis, as the reaction yields are often in the 40–90% range. Deoxynucleoside kinase, however, often outperforms deoxycytidine kinase and accepts a wider range of nucleoside analogues as substrates. Hence, deoxynucleoside kinase and deoxycytidine kinase were active towards 43 and 34 of 57 tested compounds, respectively. Both nucleoside kinases have been also tested for a larger-scale synthesis of nucleoside monophosphates in the presence of a GTP regeneration system using acetate kinase from *E. coli*.

**Keywords:** nucleoside 5′-monophosphates; modified nucleotides; deoxynucleoside kinase; deoxycytidine kinase; acetate kinase; enzymatic synthesis; biocatalysis

## 1. Introduction

Nucleoside and nucleotide analogues are an important class of chemical compounds that have a wide variety of applications. Nucleosides and their 5′-monophosphates are often used as precursors for chemical or enzymatic nucleoside 5′-triphosphate synthesis [1]. These compounds are also employed for numerous biochemical implementations where they are being used as substrates or inhibitors for certain enzymes such as polymerases or kinases. Moreover, fluorescently or radiolabeled nucleotides act as probes in nucleic acid research [2–4]. It is known that certain nucleosides and nucleotides, such as azidothymidine or gemcitabine, act as antiviral and anticancer agents [5,6].

Scientific community has spent a great deal of time and energy into developing efficient methods for nucleoside 5′-phosphate synthesis. However, a universal method, applicable for all nucleosides and their analogues is still yet to be found. Generally, modified nucleoside 5′-mono- and 5′-polyphosphates are prepared by chemical synthesis, which has its own drawbacks. Chemical nucleoside phosphorylation is often achieved by using phosphochloride agents such as phosphorylchloride or salicylic acid phosphorochloridate cyclic anhydride [1,7,8]. Such compounds are extremely reactive, hygroscopic, and harmful to people and the environment. Moreover, chemical nucleoside phosphorylation occasionally results in average or even low nucleotide yields. Chemical synthesis is further complicated by additional nucleoside functional groups such as amine or carboxy groups, which frequently require additional protection before phosphorylation and deprotection after phosphorylation.

To overcome the aforementioned hurdles, an enzymatic nucleoside phosphorylation can be employed. Nucleoside 5′-monophosphates can be synthesized by utilizing phosphorylating enzymes such as nucleoside kinases and a phosphate donor such as ATP or GTP. Deoxynucleoside kinase (dNK) from *Drosophila melanogaster* is a good example of such a kinase. The enzyme catalyzes a nonreversible phosphate group transfer from a nucleoside 5′-triphosphate, preferably ATP or GTP, to a nucleoside 5′-hydroxyl group (Figure 1). It is known that dNK is more active towards 2′-deoxynucleosides than their ribo-counterparts; however, various canonical and non-canonical nucleosides are its substrates [9]. The fruit fly dNK is a well-studied kinase, which, due to its fast turnover rate and broad substrate specificity, is commonly employed for phosphorylation of numerous nucleoside analogues. The wild-type dNK has been employed as a biocatalyst for the synthesis of various unnatural mononucleotides at both exploratory and preparatory scale with yields reaching close to 100%. The enzyme is active towards nucleosides that are based on the isocarbostyryl or 7-azaindole rings as well as multiple prodrugs including vidarabine and fludarabine [10,11]. There are also reports of dNK mutant variants that possess the capacity to phosphorylate synthetic 2′-deoxynucleosides such as dP, dZ, dX, and dK (2-amino-8-(1′-β-D-2′-deoxyribofuranosyl)-imidazo [1,2-*a*]-1,3,5-triazin-4(8*H*)-one, 6-amino-3-(1′-β-D-2′-deoxyribofuranosyl)-5-nitro-1*H*-pyridin-2-one, 8-(1′-β-D-2′-deoxy-ribofuranosyl)imidazo [1,2-*a*]-1,3,5-triazine-2(8*H*)-4(3*H*)-dione, and 2,4-diamino-5-(1′-β-D-2′-deoxyribofuranosyl)-pyrimidine, respectively) or mutants that become more specific towards purine instead of pyrimidine nucleosides, and even have the ability to phosphorylate 2′-deoxyribose sugars without a base attached [12,13]. Another commonly utilized enzyme for nucleoside phosphorylation is *Homo sapiens* deoxycytidine kinase (*Hs*dCK). In addition to natural nucleosides such as deoxycytidine, deoxyadenosine, and deoxyguanosine, *Hs*dCK phosphorylates an assortment of antineoplastic or antiviral nucleoside analogues [14,15]. *Hs*dCK has been recently applied for the synthesis of nucleoside monophosphate analogues, where an immobilized *Hs*dCK phosphorylated certain prodrugs, including cladribine, fludarabine, azadine, gemcitabine, cytarabine, lamivudine, and clofarabine with reaction yields lower than obtained using a soluble enzyme, but still reaching 20–60% with most substrates [16]. An immobilized form of deoxyadenosine kinase (*Dd*dAK), isolated from *Dyctiostelium discoideum*, a cellular slime mold, has also been employed for an enzymatic synthesis of adenine 5′-arabinonucleotides [17]. This kinase outperforms the aforementioned dNK and *Hs*dCK, as conversion of vidarabine and fludarabine into their corresponding monophosphates were as efficient as 99%. Deoxycytidine kinase (dCK) from *Bacillus subtilis* is another example of such enzymes. The *B. subtilis* kinase is less studied than the nucleoside kinases from *D. melanogaster* or *H. sapiens*; however, the available data demonstrate that dCK accepts 2′-deoxycytidine, 2′-deoxyadenosine, and 2,6-diaminopurine 2′-deoxynucleoside as substrates for phosphorylation [18,19]. Regarding a phosphate donor, dCK accepts both GTP and ATP, the same as dNK.

**Figure 1.** Nucleoside kinase catalyzed phosphorylation reaction.

The aim of this work was therefore to study and compare substrate specificities of *D. melanogaster* deoxynucleoside kinase and *B. subtilis* deoxycytidine kinase and to employ the enzymes for a large-scale synthesis of certain nucleoside 5′-monophosphates using a GTP regeneration performed by *E. coli* acetate kinase (ACK).

## 2. Results

### 2.1. Expression of Target Genes and Catalytic Verification of the Purified Enzymes

The purification of the target proteins is described in Section 4.5. All three enzymes were purified with $(His)_6$-tags at the N-termini (Figure S1, in the Supplementary Materials) and used without removal of the tags.

To verify the catalytic capabilities of the purified nucleoside kinases, enzymatic 2′-deoxycytidine 5′-monophosphate synthesis was performed using *D. melanogaster* deoxynucleoside kinase (dNK) and *B. subtilis* deoxycytidine kinase (dCK). Reaction conditions were based on experiments previously conducted on dNK and dCK enzymes [9,18]. A standard 50 mM potassium phosphate buffer with a pH value of 7.5 was chosen as a reaction medium for both kinases. Because $Mg^{2+}$ and other divalent metal ions are essential for the enzymatic phosphorylation, 15 mM of $MgCl_2$ was added to the reaction mixture, which already contained 10 mM of 2′-deoxycytidine and 15 mM of GTP [20]. To initiate the phosphorylation, 1 μL of purified kinase was added to 50 μL reaction mixtures, giving a final concentration of 0.69 nmol/mL for dNK and 9.45 nmol/mL for dCK. After undergoing the reactions for 1 h at 37 °C and 500 rpm, TLC and HPLC-MS analyses were performed. The analyses confirmed that both kinases successfully phosphorylated 2′-deoxycytidine using GTP as a phosphate donor with high yields (Figure S2, in the Supplementary Materials), giving specific activities of 1.6 and 35.9 μmol min$^{-1}$ mg$^{-1}$ for dCK and dNK, respectively.

### 2.2. Optimization of Reactions Catalyzed by dNK and dCK

To find the optimal nucleoside phosphorylation reaction conditions, an impact of multiple parameters, such as reaction duration, temperature, medium pH, nucleoside concentration, GTP concentration, and enzyme concentration, was assessed using HPLC-MS chromatogram area sizes, and the results between dNK and dCK were compared. Unless stated otherwise, reaction mixtures contained 10 mM dCyd, 15 mM GTP, 15 mM $MgCl_2$, and 0.69 nmol/mL dNK or 9.45 nmol/mL dCK in 50 mM potassium phosphate buffer (pH 7.5) and were performed at 37 °C at 500 rpm for 5 min.

The optimal reaction duration was evaluated by applying an appropriate kinase (the final concentrations of enzymes were 0.69 nmol/mL and 9.45 nmol/mL for dNK and dCK, respectively) in a 50 μL reaction volume for 1–120 min. As can be seen from the time dependency chart in Figure 2a, both reactions proceeded in a similar fashion and started plateauing after 30 min, reaching a maximum dCMP yield of nearly 100% 1 h after starting.

The effect of temperature on the enzymatic conversion of deoxycytidine to deoxycytidine monophosphate was measured at different temperatures, ranging from 4 to 80 °C. It was determined that both enzymes possessed similar characteristics and retained catalytic activity in the entire range of the tested temperatures (Figure 2b). Both dNK and dCK had the lowest activities at the 4–20 °C range and yields of nucleotide reached 10–20% after 5 min. The catalytic efficiencies of both enzymes rose until 60 °C. Under these conditions, the yield of monophosphate was almost 100% after 5 min of reaction. *D. melanogaster* dNK demonstrated a lower sensitivity to high temperature compared to *B. subtilis* dCK; hence, phosphorylation of deoxycytidine at 70 °C proceeded as good as at 60 °C. Catalytic activities of both kinases fell dramatically when reaction temperatures were held at 80 °C: dNK phosphorylation efficiency went from almost 100% to 70%, and dCK efficiency plunged to 20%.

The impact of the pH of the reaction medium for dNK and dCK activity was evaluated using potassium phosphate (pH 5.5–7.0) and tris–HCl (pH 7.0–9.0) buffers. Reaction mixtures containing dNK reached the highest deoxycytidine monophosphate yield (50%) in the 5 min timeframe when reaction medium had a pH value of 8.0, although similar yields can be obtained in the pH range of 7.5–8.5. It is evident from Figure 2c that dNK activity dropped drastically when the pH reached 9.0. Kinase dCK possesses similar properties to dNK. The enzyme reached the greatest efficiency at pH 8.5, although similar deoxycytidine monophosphate yields as high as 50% can be achieved using a reaction medium pH that ranges from 8.0 to 9.0.

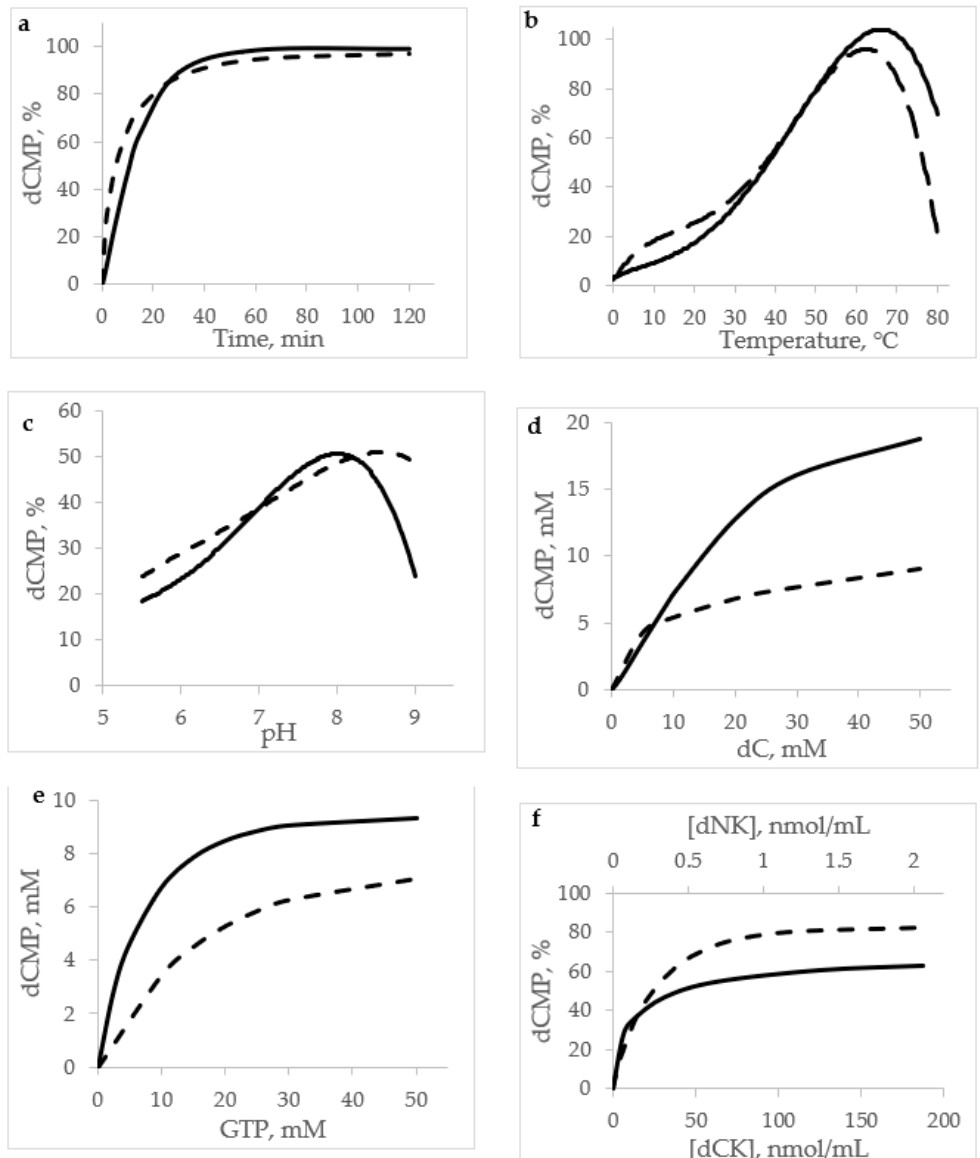

**Figure 2.** Evaluation of various parameters on dNK and dCK activity towards dCyd. The solid line represents dNK and the dashed line represents dCK. (**a**) Influence of reaction duration on the reaction yield; (**b**) dependency of product yield on reaction temperature; (**c**) an impact of pH on dNK and dCK activity; (**d**) effect of initial substrate concentration on the reaction yield; (**e**) effect of GTP concentration on the reaction yield; (**f**) impact of kinase concentration on the reaction yield.

The impact of nucleoside concentration was measured using 1–50 mM of deoxycytidine and a final concentration of dNK and dCK of 0.69 nmol/mL (34 pmol) and 9.45 nmol/mL (470 pmol), respectively. As depicted in Figure 2d, in the 5 min timeframe 34 pmol of dNK was capable of converting up to 18 mM of dCyd (amounting to 26.2 pmol of dCyd converted by 1 pmol of dNK), while 470 pmol of dCK could only synthesize around 8 mM of deoxycytidine monophosphate (amounting to 0.85 pmol of dCyd converted by 1 pmol of dNK), making dNK around 30 times faster than dCK. Nucleoside concentrations of up to 50 mM were well tolerated by the kinases and no inhibition was observed.

Evaluation of phosphate donor concentration revealed that to run the phosphorylation reaction of dCyd optimally in the presence of both dNK and dCK, at least 1.5 molar equivalent of GTP should be used in the system. A further increase in concentration of GTP had only a miniscule effect on the yield of dCMP (Figure 2e).

Optimal concentrations of dNK and dCK in 50 μL reaction volumes were evaluated using varying enzyme concentrations (0.07 to 2.06 nmol/mL for dNK and 1.89 to 189 nmol/mL for dCK). As can be seen from Figure 2f, 0.34 nmol/mL of dNK, which equates to 0.5 μL of enzyme solution in 50 μL reaction mixture, is the point where dCMP formation starts to taper off. It was determined that dCK is a slower enzyme than dNK, and 5 μL of enzyme solution in 50 μL reaction mixture (47 nmol/mL) is necessary for the synthesis to proceed at the optimal speed.

It was determined that dCyd phosphorylation performed by both dNK and dCK took up to an hour when using 10 mM dCyd, 15 mM GTP, and 0.69 nmol/mL dNK or 9.45 nmol/mL dCK. The highest catalytic efficiencies were reached when temperatures were 60 °C for dCK and 70 °C for dNK. A reaction medium pH of 8.0 was optimal for dNK, and a pH of 8.5 was optimal for dCK. However, due to the sensitive nature of synthesized mononucleotides and potential denaturation of enzymes during prolonged reactions, a temperature of 37 °C and a pH 7.5 were chosen for further studies. Both kinases tolerated dCyd concentrations that reached up to 50 mM well, with no detected inhibition. It was established that for the reactions to run optimally, concentrations of 0.34 nmol/mL of dNK or 47 nmol/mL of dCK were necessary. As for the phosphate donor, 1.5 molar equivalent of GTP should be used in the reaction mixtures.

### 2.3. Substrate Specificity of the Nucleoside Kinases

The substrate specificity of *D. melanogaster* dNK and *B. subtilis* dCK were tested using 57 various canonical and modified nucleosides, 44 of which were suitable substrates to at least one of the nucleoside kinases (Tables 1 and 2). Reaction conditions were based on the results of reaction optimization assays described in Section 2.2. Nucleoside concentration was chosen to be 10 mM while using 1.5 mol. eq. (15 mM) of GTP and MgCl$_2$. Because the reactions were executed for 24 h, 1 μL of each kinase was used in a reaction volume of 50 μL, which corresponds to 0.69 nmol/mL for dNK and 9.45 nmol/mL for dCK. Although greater enzymatic activities were achieved using pH 8.0–8.5 and temperature of 50–60 °C, 50 mM potassium phosphate buffer with a pH value of 7.5 and a temperature of 37 °C were chosen due to the sensitive nature of utilized nucleosides and their monophosphates.

**Table 1.** Efficiency of synthesis of canonical nucleoside monophosphates and cytidine derivative monophosphates by dNK and dCK. HPLC-MS analysis of the reaction mixtures was performed after incubation at 37 °C for 24 h. nd—the nucleoside monophosphate was not detected by HPLC-MS.

| Nucleoside | No. | R$_1$ | R$_2$ | Conversion, % | |
| --- | --- | --- | --- | --- | --- |
| | | | | dNK | dCK |
| | 1 | (cytosine) | -H | 87 | 92 |
| | 2 | (cytosine) | -OH | 84 | 95 |
| | 3 | (uracil) | -H | 61 | 97 |

**Table 1.** *Cont.*

| Nucleoside | No. | R$_1$ | R$_2$ | Conversion, % | |
| --- | --- | --- | --- | --- | --- |
| | | | | dNK | dCK |
| | **4** | | -OH | 78 | 97 |
| | **5** | | -H | 64 | 31 |
| | **6** | | -H | 94 | 98 |
| | **7** | | -OH | 21 | 43 |
| | **8** | | -H | 26 | 88 |
| | **9** | | -OH | 35 | 63 |
| | **10** | | -OH | 62 | 7 |
| | **11** | | -H | 82 | 37 |

**Table 1.** *Cont.*

| Nucleoside | No. | R₁ | R₂ | Conversion, % | |
|---|---|---|---|---|---|
| | | | | dNK | dCK |
|  | **12** | -F | - | 8 | 98 |
| | **13** | -CH₃ | - | 22 | 50 |
|  | **14** |  | -H | 34 | 95 |
| | **15** |  | -H | 10 | 32 |
| | **16** | -OH | -OH | 60 | 38 |
| | **17** |  | -H | nd | 37 |
| | **18** |  | -H | nd | nd |
| | **19** |  | -H | 4 | nd |
| | **20** |  | -H | 2 | nd |
| | **21** |  | -H | 12 | nd |
| | **22** |  | -H | 11 | nd |
| | **23** |  | -H | nd | nd |

**Table 1.** *Cont.*

| Nucleoside | No. | R₁ | R₂ | Conversion, % | |
|---|---|---|---|---|---|
| | | | | **dNK** | **dCK** |
|  | 24 |  | -H | nd | nd |
| | 25 |  | -H | nd | nd |
| | 26 |  | -H | nd | nd |
| | 27 |  | -H | 93 | 34 |
| | 28 |  | -H | 5 | 27 |
| | 29 |  | -H | 11 | 16 |
| | 30 |  | -H | 74 | 60 |
| | 31 |  | -H | nd | nd |
| | 32 |  | -H | nd | nd |

**Table 2.** Efficiency of synthesis uridine derivative monophosphates by dNK and dCK. HPLC-MS analysis of the reaction mixtures was performed after incubation at 37 °C for 24 h. nd—the nucleoside monophosphate was not detected by HPLC-MS.

| Nucleoside | No. | $R_1$ | $R_2$ | $R_3$ | Conversion, % | |
|---|---|---|---|---|---|---|
| | | | | | dNK | dCK |
|  | 33 | -H | - | - | 80 | 44 |
| | 20 | -CH$_3$ | - | - | 40 | 93 |
|  | 35 | -S | -O | - | 91 | 98 |
| | 36 | -O | -S | - | 49 | 88 |
|  | 37 | -CH$_3$ | -F | - | 29 | 8 |
| | 38 | -CH$_3$ | -H | - | 57 | 11 |
| | 39 | -CH$_2$CH$_3$ | -F | - | 0.4 | nd |
| | 40 | -CH$_2$CH$_3$ | -H | - | 70 | nd |
| | 41 |  | -F | - | nd | nd |
| | 42 |  | -H | - | nd | nd |
|  | 43 | -F | -OH | - | nd | nd |
| | 44 | -H | -H | - | nd | nd |
|  | 45 | -OH | -O | - | 94 | 99 |
| | 46 | -H | -S | - | 49 | nd |

**Table 2.** *Cont.*

| Nucleoside | No. | R$_1$ | R$_2$ | R$_3$ | Conversion, % | |
|---|---|---|---|---|---|---|
| | | | | | dNK | dCK |
| | 47 | -OCH$_3$ | -OH | -H | 24 | 17 |
| | 48 | -OH | -OCH$_3$ | -H | 16 | nd |
| | 49 | -OCH$_3$ | -OH | -CH$_3$ | 21 | nd |
| | 50 | -OCH$_2$CHCH$_2$ | -OH | -H | 14 | 21 |
| | 51 | -OH | -OCH$_2$CHCH$_2$ | -H | 39 | nd |
|  | 52 | (cyclic acetonide across R$_1$/R$_2$) | | -H | 14 | 70 |
| | 53 | -OH | (acetyl ester) | -H | nd | nd |
| | 54 | -H | (benzoyl ester) | -H | nd | nd |
| | 55 | -NH$_2$ | -OH | -H | 56 | 40 |
| | 56 | (N-acetylamino) | -OH | -H | 32 | 54 |
| | 57 | -H | -H | -H | 65 | 29 |

Previous studies have established that in addition to deoxycytidine, *B. subtilis* dCK possesses specificity towards deoxyadenosine [21]. *D. melanogaster* dNK, on the other hand, is more active towards pyrimidine nucleosides, although purine nucleosides can also be its substrates [22]. Both kinases are more active towards 2′-deoxynucleosides than their ribo-equivalents. Our investigation has confirmed that both kinases phosphorylated all canonical 2′-deoxy- and ribonucleosides, albeit with slight differences in substrate specificity (Table 1, **1–9**). As expected, dNK phosphorylated pyrimidine nucleosides, such as 2′-deoxycytidine, cytidine, 2′-deoxyuridine, uridine, and thymidine with conversion efficiencies reaching 95%. Meanwhile purine nucleosides, such as guanosine, 2′-deoxyadenosine, and adenosine were phosphorylated to the corresponding 5′-monophosphates with yields that reached up to 35% (with the exception of 2′-deoxyguanosine, which was phosphorylated as great as pyrimidine nucleosides). Deoxycytidine kinase showed similar phosphorylation efficiencies to previously mentioned pyrimidine nucleosides (apart from thymidine, which was converted with 31% efficiency) and superior phosphorylation yields to the aforementioned purine nucleosides compared to dNK: 2′-deoxyguanosine, guanosine, 2′-deoxyadenosine and adenosine phosphorylation yields were between 43% and 98%. Both dNK and dCK were capable of phosphorylating ribo- and 2′-deoxyisocytidine and 5-methyl- and 5-fluorocytidine. The two latter compounds were phosphorylated by dCK with high yields (>50%), whereas modifications at C5 position affected dNK negatively: the yields barely reached 8–22%. On the other hand, compared to dCK, dNK was more efficient at phosphorylating ribo- and 2′-deoxynucleosides with isocytosine nucleobase.

Application of cytidine nucleosides with modifications at N$^4$ position returned mixed results (Table 1, **14–32**). Some of the compounds were better suited to be substrates to dNK, others were better phosphorylated by dCK, and some were not suitable for either enzyme. Both kinases accepted nucleosides with small modifications at N$^4$ position: dNK phosphorylated N$^4$-acetyl-2′-deoxycytidine with a 34% yield and dCK converted the

compound to a monophosphate almost completely (95%). Structurally comparable $N^4$-isobutyryl-2′-deoxycytidine (**15**) was also an acceptable substrate for both kinases; however, the reaction yields were lower in comparison to the yields of $N^4$-acetyl-2′-deoxycytidine 5′-monophosphate and topped out at 10% for dNK and 32% for dCK. Cytidine with a hydroxyl group at the $N^4$ position (**16**) was a better substrate to dNK compared to dCK, although both kinases phosphorylated the compound with moderate yields that reached no more than 38%.

*D. melanogaster* dNK was unable to phosphorylate N-(1-((2R,4S,5R)-4-hydroxy-5-(hydroxymethyl)tetrahydrofuran-2-yl)-2-oxo-1,2-dihydropyrimidin-4-yl)isonicotinamide (**17**) when dCK phosphorylated the nucleoside with an almost 40% reaction yield. Interestingly, a very similar compound $N^4$-benzoyl-2′-deoxycytidine (**18**), whose only difference from the beforementioned nucleoside was an absence of a nitrogen atom at the benzene ring, was not phosphorylated by either kinase. When it came to our previously synthesized $N^4$ amino acid-modified 2′-deoxycytidines, dNK was superior to dCK, and phosphorylated deoxycytidines that were modified with glycine and alanine amino acids both with and without a Boc protecting group (**19–22**), even though the phosphorylation yields were low (2–12%) [23]. Nucleosides with bulkier amino acid residues, such as 2-((1-((2R,4S,5R)-4-hydroxy-5-(hydroxymethyl)tetrahydrofuran-2-yl)-2-oxo-1,2-dihydropyrimidin-4-yl)amino)-3-(4-hydroxyphenyl)propanamide (**23**), tert-butyl (1-((1-((2R,4S,5R)-4-hydroxy-5-(hydroxymethyl)tetrahydrofuran-2-yl)-2-oxo-1,2-dihydropyrimidin-4-yl)amino)-3-(4-hydroxyphenyl)-1-oxopropan-2-yl)carbamate (**24**), tert-butyl (1-((1-((2R,4S,5R)-4-hydroxy-5-(hydroxymethyl)tetrahydrofuran-2-yl)-2-oxo-1,2-dihydropyrimidin-4-yl)amino)-4-methyl-1-oxopentan-2-yl)carbamate (**25**), and tert-butyl (1-((1-((2R,4S,5R)-4-hydroxy-5-(hydroxymethyl)tetrahydrofuran-2-yl)-2-oxo-1,2-dihydropyrimidin-4-yl)amino)-3-(1H-indol-3-yl)-1-oxopropan-2-yl)carbamate (**26**) were not phosphorylated by the nucleoside kinases.

*D. melanogaster* dNK was superior to *B. subtilis* dCK at phosphorylating deoxycytidine with a sec-butyl group at $N^4$-position (**27**), as the nucleoside conversion exceeded 90%, whereas the nucleotide yield by dCK reached only 34%. Contrary to the previous results, dCK surpassed dNK at phosphorylating bulkier 2′-deoxycytidines with phenylethyl modifications at $N^4$ position (**28** and **29**): dCK reaction yields were 16–27%, whereas dNK reaction yields only reached 5–11%. Deoxycytidine with a phenyl modification at $N^4$ position (**30**), however, was converted to a nucleotide by the kinases very similarly: nucleotide yields were above average and reached 60–74%. Neither 2′-deoxycytidine modified with D-glucose at $N^4$ position (**31**) nor $N^4$-decyl-2′-deoxycytidine (**32**) were phosphorylated by the kinases.

Both kinases proved to be effective at phosphorylating a handful of ribo- and 2′-deoxyuridine derivatives (Table 2). The pair of kinases successfully converted pseudouridine (**33**), $N^1$-methylpseudouridine (**34**), 2-thiouridine (**35**), and 4-thiouridine (**36**) to the corresponding 5′-monophosphates with reaction yields between 40% and 98%. The enzymes accepted multiple thiouridines as substrates: dCK managed to phosphorylate 5-fluoro-4-methylthiouridine (**37**) and 4-methylthiouridine (**38**) with reaction yields reaching 10%, but analogues, containing an ethyl group instead of methyl (**39** and **40**), were not accepted as substrates. All of the aforementioned compounds were phosphorylated by dNK with moderate yields (29–70%), although 5-fluoro-4-ethylthiouridine 5′-monophosphate formation was miniscule (0.4%). Bulkier thiouridine derivatives were not suitable substrates for the nucleoside kinases: 5-fluoro-4-benzylthiouridine (**41**) and 4-benzylthiouridine (**42**) monophosphate formation was not detected. As to be expected, 5-fluoro-5′-deoxyuridine (**43**) and 2′,5′-dideoxyuridine (**44**) were not phosphorylated by dNK or dCK, as there are no 5′-hydroxyl groups to attach a phosphate to. Similar to previous results, a hydroxyl group at the nucleobase had no negative impact for reaction yields: 5-hydroxy-2′-deoxyuridine (**45**) was converted to a nucleotide almost completely by both kinases. Oddly, dCK was unable to phosphorylate 2′-deoxy-4-thiouridine (**46**), whereas 4-thiouridine was accepted as a substrate by the kinase. Both nucleosides were successfully phosphorylated by dNK with moderate reaction yields.

Interestingly, modifications at the nucleoside ribose were slightly better tolerated by dNK. The kinase from *D. melanogaster* was able to utilize uridines containing methyl and allyl groups at ribose 2′- and 3′-hydroxy groups (**47–51**) as substrates, even though the yields were low to moderate (14–39%), whereas dCK managed to phosphorylate only 2′-O-methyluridine (**47**) and 2′-(O-allyl)-uridine (**50**). Compound **52**, 2′,3′-O-isopropylidineuridine, which had modifications at both 2′-OH and 3′-OH, was an outlier: both kinases phosphorylated the compound, although dCK did it with almost five times the monophosphate yield (70%). Phosphorylation of uridine derivatives with bulkier modifications at the ribose 3′ position did not yield positive results: 3′-O-acetyluridine (**53**) and 3′-O-benzoyl-2′-deoxyuridine (**54**) were not accepted as substrates by either nucleoside kinase. Uridines, with smaller modifications at ribose 2′ position, such as 2′-amino-2′-deoxyuridine (**55**) and 2′-N-acetyl-2′-amino-2′-deoxyuridine (**56**), as well as 2′,3′-dideoxyuridine (**57**), were well tolerated by the kinases: nucleotide yields were between 29% and 65%.

*2.4. Larger-Scale Syntheses of Nucleoside 5′-Monophosphates*

Larger-scale synthesis of mononucleotides was set up using the results obtained from the phosphorylation optimization assay described in Section 2.2. Using an enzymatic cascade system (Figure 3) that consisted of a nucleoside kinase and an acetate kinase, three mononucleotides were synthesized: 2′-deoxycytidine 5′-monophosphate and $N^4$-acetyl-2′-deoxycytidine 5′-monophosphate employing dCK, and 2-thiouridine 5′-monophosphate employing dNK, utilizing GTP as a phosphate donor. Because during the reaction GTP is exhausted and turned into GDP, which cannot be further used as a phosphate donor, a GTP regeneration system was added. The regeneration is performed by acetate kinase, which transfers a phosphate group from acetyl phosphate to a GDP molecule, forming GTP and acetic acid [24]. This allows for the downscaling of the amount of GTP used in the reaction mixture, which not only makes the reaction less expensive, but also simplifies the mononucleotide purification process.

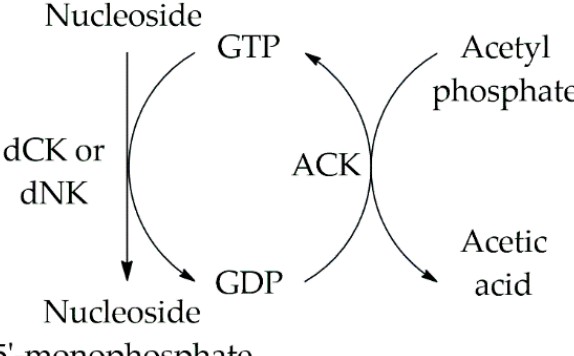

**Figure 3.** Nucleoside 5′-monophosphate synthesis using an enzymatic cascade that consists of *D. melanogaster* dNK or *B. subtilis* dCK and *E. coli* ACK.

The syntheses were performed using 50 mM concentrations of nucleosides in 50 mM potassium phosphate buffer (pH 7.5), which amounted to 300 mg of 2′-deoxycytidine, 161 mg of $N^4$-acetyl-2′-deoxycytidine, and 200 mg of 2-thiouridine. Deoxycytidine kinase-phosphorylated 2′-deoxycytidine and $N^4$-acetyl-2′-deoxycytidine, and deoxynucleoside kinase were chosen for 2-thiouridine 5′-monophosphate synthesis. Because the phosphorylation reactions were supplemented with a GTP regeneration system, only 5 mM of GTP was added to reaction mixtures. To ensure that the GTP was sufficiently regenerated, 100 mM of acetylphosphate was added to the reaction mixtures every 24 h. Since the aqueous solution of acetylphosphate is most stable at pH 5–6 and −35 °C, and at neutral pH and room temperature, the half-life of the compound is 21 h, recurrent supplementation was necessary [25,26]. Due to the diminishing activities of the utilized enzymes, in addition to the initial enzyme concentrations of 0.34 nmol/mL of dNK or 9.45 nmol/mL

of dCK and 0.46 nmol/mL of ACK, the mixtures were supplemented with the same initial amounts of each enzyme every 24 h. After carrying out the reactions for 72 h and purifying the synthesized monophosphates with ion-exchange and reverse-phase chromatographies, the structures of the compounds were confirmed by HPLC-MS and NMR analyses. The obtained amounts of the nucleotides were calculated using the UV spectra of their corresponding nucleosides, and were as follows: 17 mg of 2′-deoxycytidine (4% final yield), 11 mg of N$^4$-acetyl-2′-deoxycytidine (5% final yield), and 16 mg of 2-thiouridine (6% final yield).

## 3. Discussion

Enzymatic nucleoside 5′-monophosphate synthesis is a viable alternative to the commonly employed chemical nucleoside phosphorylation methods. Our work has proven that enzymatic phosphorylation by nucleoside kinases can potentially reach the same, if not higher mononucleotide reaction yields. Moreover, the reactions catalyzed by kinases are relatively quick, environmentally conscious, and rather inexpensive, especially when utilizing a GTP regeneration system performed by acetate kinase. The greatest advantage of enzymatic phosphorylation, however, is that nucleosides with various modifications can be used as substrates without the need of protecting their functional groups.

The nucleoside kinases chosen for our work can phosphorylate all canonical nucleosides and numerous nucleoside analogues with modifications at nucleobases or sugars. *B. subtilis* dCK is more active towards cytosine, uracil, and adenine 2′-deoxynucleosides and accepts analogues with minor modifications at the nucleobases, such as thio, acetyl, fluoro, hydroxy, or methyl groups. *D. melanogaster* dNK is a nonspecific enzyme that possesses activity towards pyrimidine 2′-deoxynucleosides, although purine nucleosides were proven to be suitable substrates as well. The kinase phosphorylated nearly all tested nucleosides that were accepted as substrates by deoxycytidine kinase, and considerably more. *D. melanogaster* kinase accepted nucleosides with bulkier modifications at nucleobases, such as 2′-deoxycytidine and tyrosine conjugate (Table 1, **16**) or 4-ethylthiouridine (Table 2, **40**). In addition to bulkier nucleobase modifications, dNK was more tolerant of modifications at nucleoside ribose. Whereas dCK could only phosphorylate nucleosides with minor modifications, such as methyl or allyl, at the ribose 2′-position, dNK accepted substrates with beforementioned modifications at 3′-position as well. However, uridines with bulkier sugar modifications, such as acetyl or benzoyl, at the ribose 3′-position were not suitable substrates for the kinase. Because the study was mostly focused on nucleoside analogues with modifications at the nucleobases, it remains unknown whether the chosen kinases would accept nucleoside analogues with unconventional sugars, such as galactose or hexose, instead of ribose. Based on our limited experiments with sugar-modified nucleosides and available literature on *D. melanogaster* dNK, the enzyme or its mutant variant might phosphorylate such substrates [12,27].

The differences between the substrate specificities of the nucleoside kinases can be explained by their active site structures. The crystal structure of a dNK–dCyd complex is known and well documented [28]. The active site of the kinase is an elongated cavity that possesses an abundance of hydrophobic residues on its top and bottom. When inside the cavity, the substrate interacts with polar residues (Figure 4). The 5′-OH of ribose moiety forms hydrogen bonds with Glu52 and Arg169, and the 3′ oxygen atom interacts with Tyr70 and Gln172. The nucleobase is stabilized by Phe114, Trp57, and Phe80 residues. The cytosine nucleobase forms two hydrogen bonds with Gln81, and the oxygen atom at C2-position of pyrimidine ring interacts with two water molecules. The wide range of accepted nucleosides by dNK can be explained by an empty space near the 5-position of the pyrimidine nucleobase lined with hydrophobic Val84, Met88, and Ala110 residues, which enables accommodation of nucleosides with modifications at pyrimidine nucleobase 4- and 5-positions. The crystal structure of dCK, on the contrary, is not determined. The kinase is structurally similar to human dCK (sequence identity 25%), and the active site is comprised of corresponding residues [29,30]. The residues with which the nucleoside interacts in

the active center of dCK are the same as those in the dNK. Residues Gln63, Arg70, and Asp93 are responsible for anchoring the nucleobase of 2′-deoxycytidine, and the 5′-OH is hydrogen bonded to Arg88. The main difference between these two kinases is that the active site of dCK does not possess an empty pocket as dNK does, thus the spectrum of accepted substrates by dCK is more limited.

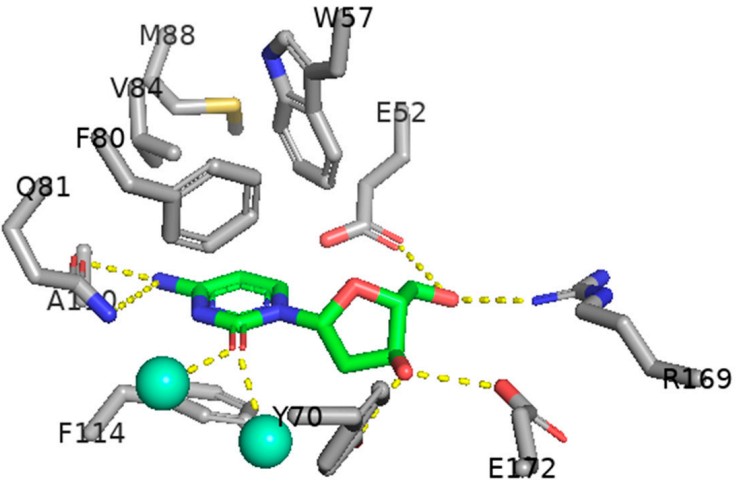

**Figure 4.** Close-up view of dCyd bound at the active site of dNK. Hydrogen bonds are yellow dashed lines; water molecules are cyan spheres. PDB ID: 1J90.

Although the yields of larger-scale synthesis of nucleoside 5′-monophosphates were below 10%, small-scale synthesis has proven that it is possible to achieve nucleoside conversion yields in the 50–99% range using the tested kinases. The low mononucleotide yields reported in this study were a result of several causes. First, acetic acid is formed due to acetylphosphate being exhausted by ACK and by natural breakdown, both of which in turn lower the pH of the reaction mixture. A higher molarity buffer might fix this issue. Second, when larger reaction volumes were adopted, enzyme precipitation was observed, especially when automatic mixing was applied. A gentler approach, such as manual mixing, partially solves the problem. Third, alternative purification methods for the mononucleotides should be addressed. The compounds synthesized in this study were purified twice using ion-exchange and reverse-phase chromatography. Because the synthesized nucleotides have only one phosphate group, the retention of the compounds in the purification columns is poor and separation from impurities is complicated, which consequently causes an unavoidable partial loss of the products. Given that additional optimization reactions for a larger scale are performed and higher mononucleotide yields are obtained, the compounds could be further employed for enzymatic nucleoside 5′-triphosphate synthesis. To achieve this, additional kinases, such as mononucleotide kinase and dinucleotide kinase, should be added to the enzymatic cascade. The available data show that mononucleotide kinases are somewhat strict in their preference of substrates; thus, suitable enzyme selection should be made depending on the chosen substrates [31]. For example, bacterial CMP kinase accepts CMP, dCMP, and arabinofuranosyl-CMP [32]. Viral thymidylate kinase, in addition to thymidylate, phosphorylates uridine and guanosine deoxymononucleotides [33]. To obtain a final trinucleotide form, a nucleoside diphosphate kinase (NDPK) would be required. NDPKs are found in prokaryotes and eukaryotes, and are considered "non-specific" enzymes due to a wide substrate cleft that allows for broad substrate specificity [34,35]. For example, human NDPK phosphorylates all canonical nucleoside diphosphates and many nucleotide analogues, such as ribavirin or azidothymidine diphosphates [36].

## 4. Materials and Methods

### 4.1. General Information

Chemicals and solvents were purchased from Sigma-Aldrich (Darmstadt, Germany), Alfa Aesar (Kandel, Germany), Honeywell (Machelen, Belgium), Fluka (Buchs, Switzerland), Merck (Vilnius, Lithuania) and GE Healthcare (Freiburg, Germany), and were of analytical grade or higher. A full list of nucleosides utilized for the study is provided in the Supplementary Data. Thin-layer chromatography (TLC) was performed on 25 TLC aluminum sheets coated with silica gel 60 F254 (Merck) and reverse-phase chromatography was performed on Grace flash cartridges C-18 (Fisher Scientific, Loughborough, United Kingdom). Ion exchange chromatography was performed on DEAE Sephadex A-25 (GE Healthcare). NMR spectra were recorded in DMSO-$d_6$ on a Bruker Ascend 400 (Ettlingen, Germany) $^1$H NMR–400 MHz, $^{13}$C NMR–101 MHz, $^{31}$P NMR–202 MHz. Chemical shifts are reported in ppm relative to the solvent resonance signal as an internal standard. UV spectra were recorded on a Lambda 25 Perkin Elmer UV/VIS spectrometer (Beaconsfield, United Kingdom). HPLC-MS analyses were performed using a high-performance liquid chromatography system equipped with a photo diode array detector (SPD-M20A, Shimadzu, Kyoto, Japan) and a mass spectrometer (LCMS-2020, Shimadzu, Kyoto, Japan) equipped with an ESI source. The chromatographic separation was conducted using a YMC Pack Pro column, 3 × 150 mm (YMC, Kyoto, Japan) at 40 °C and a mobile phase that consisted of 0.1 formic acid water solution (solvent A) and acetonitrile (solvent B). Mass spectrometry data were acquired in both positive and negative ionization modes and analyzed using the LabSolutions LCMS software (version 5. 42 SP6). The close-up view of dNK–dCyd complex was modelled using the PyMOL Molecular Graphics System, Version 2.0 Schrödinger, LLC.

### 4.2. Acetylphosphate Synthesis

Acetylphosphate synthesis was performed based on a method previously described by Crans and Whitesides [25]. A volume of 28.2 mL (0.3 mol) of cooled acetic acid anhydride was slowly poured into a chilled mixture of 6.75 mL (0.1 mol) of 85 wt.% phosphoric acid and 60 mL (0.6 mol) of ethyl acetate and the solution was mixed for 5.5 h at 0 °C. The reaction mixture was poured into a flask that contained 50 mL of $H_2O$, 25 g of ice and 8.4 g (0.1 mol) of sodium bicarbonate and was mixed until carbon dioxide gas stopped forming. The solution was poured into a separatory funnel separating the water phase from organic phase. The extracted water phase was washed three times with 100 mL of ethyl acetate and neutralized using 10 M lithium hydroxide solution. The neutralized solution was washed with 50 mL of ethyl acetate and concentrated to a final volume of 75 mL using a rotary evaporator. The final aqueous solution of 1 M acetylphosphate was obtained, and the structure of the compound was confirmed using HPLC-MS and NMR analyses.

### 4.3. Bacterial Strains, Plasmid, and Reagents

The deoxynucleoside kinase (dNK) gene was encoded by pOpen-dromedNK plasmid, which was kindly provided by Drew Endy and Jennifer Molloy and FreeGenes Project (Addgene plasmid #165579; http://n2t.net/addgene:165579; RRID:Addgene_165579; Watertown, MA, USA; accessed on 12 October 2021), the deoxycytidine kinase (dCK) gene was encoded by *Bacillus subtilis* 168 (from laboratory stock of Department of Molecular Microbiology and Biotechnology, Institute of Biochemistry, Life Sciences Center, Vilnius University, Vilnius, Lithuania), and the acetate kinase (ACK) gene was encoded by Escherichia coli DH5α (Novagen, Darmstadt, Germany). LIC cloning and expression set that contained the expression vector, pLATE31, was purchased from ThermoFisher Scientific (Vilnius, Lithuania). Escherichia coli DH5α was used for plasmid amplification and Escherichia coli BL21 (DE3) was used for the expression of the target proteins. Both strains were purchased from Novagen (Darmstadt, Germany).

### 4.4. Cloning of Target Genes

The target genes were amplified from pOpen-dromedNK plasmid (dNK), *B. subtilis* 168 (dCK) and *E. coli* DH5$\alpha$ (ACK) by PCR using Phusion polymerase with synthetic primers (Table 3). Amplified genes were purified from agarose gel and cloned into pLATE31 plasmid according to manufacturer's protocol. The constructed vectors were electroporated into *E. coli* DH5$\alpha$ electro-competent cells. Transformants were selected on LB agar plates (0.1 mg/mL ampicillin). The isolated vectors were named pLATE31-dNK, pLATE31-dCK, and pLATE31-ACK and sequenced.

**Table 3.** PCR primers used for target gene amplification.

| Gene Name | Gene Source | Primer Name | Primer Sequence 5'→3' |
|---|---|---|---|
| *dNK* | pOpen-dromedNK | dNK_F<br>dNK_R | AGAAGGAGATATAACTATGGCGGAAGCAGCAAGCTG<br>GTGGTGGTGATGGTGATGGCCGCGTGCAACACGCTGACG |
| *dCK* | *B. subtilis* 168 | dCK_F<br>dCK_R | AGAAGGAGATATAACTATGAAGGAACATCATATC<br>GTGGTGGTGATGGTGATGGCCCTTTTTTTTGATTATCATG |
| *ACK* | *E. coli* DH5$\alpha$ | ACK_F<br>ACK_R | AGAAGGAGATATAACTATGTCGAGTAAGTTAGTAC<br>GTGGTGGTGATGGTGATGGCCGGCAGTCAGGCGGCTC |

### 4.5. Expression of Target Genes and Purification of the Enzymes

The recombinant plasmids were transformed into electro-competent *E. coli* BL21 (DE3) cells. Transformants were selected on 0.1 mg/mL ampicillin LB agar plates, inoculated into two conical flasks containing 200 mL of LB medium with 0.1 mg/mL ampicillin and incubated at 37 °C with a shaking speed of 180 rpm for 2–5 h. After $OD_{600}$ reached 0.6, gene expression was induced by addition of 1 mM of IPTG and incubating for 3 h at 37 °C and 180 rpm. Cells were collected by centrifugation at 4000× *g* at 4 °C for 10 min. The supernatant was discarded, and the remaining cells were suspended in 10 mL of buffer A (50 mM potassium phosphate buffer, pH 7.5) and lysed by sonication (5 min; 2 s disruption, 8 s cooling; 22 kHz at 40% amplitude). The soluble fractions were obtained by centrifugation at 4000× *g* at 4 °C for 10 min. Cell-free lysates were loaded onto 5 mL $Ni^{2+}$ HiTrap chelating HP column (Cytiva, Marlborough, MA, USA), equilibrated with buffer A. The enzymes were eluted using buffer B (50 mM potassium phosphate, imidazole 0.5 M, pH 7.5) gradient from 0 to 100%, at a flow rate of 1 mL/min. Protein purification was monitored using 14% sodium dodecyl sulphate–polyacrylamide gel electrophoresis (SDS-PAGE). Fractions containing target proteins were pooled, placed into dialysis bags, and dialyzed overnight at 4 °C in buffer A. *D. melanogaster* dNK was diluted two times with glycerol, and dCK and ACK were concentrated with carboxymethyl cellulose. The enzymes were stored at −20 °C until further use.

### 4.6. Catalytic Verification of Purified Enzymes

To verify that the purified enzymes were functional, an activity assay was performed. The reaction conditions were as follows: 10 mM 2'-deoxycytidine, 15 mM GTP, 15 mM $MgCl_2$, 50 mM potassium phosphate buffer (pH 7.5), 0.02 mg/mL kinase (for dNK); 10 mM 2'-deoxycytidine, 15 mM GTP, 15 mM $MgCl_2$, 50 mM potassium phosphate buffer (pH 7.5), 0.24 mg/mL kinase (for dCK); and 10 mM GDP, 15 mM acetylphosphate, 40 mM $MgCl_2$, 50 mM potassium phosphate buffer (pH 7.5), 0.4 mg/mL kinase (for ACK). The reactions were performed for 1 h at 37 °C and 500 rpm. Deoxycytidine monophosphate formation was qualitatively detected with TLC and HPLC-MS analyses.

### 4.7. Optimization of Reactions Catalyzed by dNK and dCK

Before the determination of nucleoside kinase substrate specificities, the effects of various factors such as reaction duration, temperature, pH, substrate, and enzyme concentrations were evaluated. The influence of these factors was quantitatively determined by HPLC-MS analysis based on the generation of 2'-deoxycytidine 5'-monophosphate (dCMP) from 2'-deoxycytidine (dCyd).

### 4.7.1. Optimal Reaction Duration

The conversion of 2′-deoxycytidine was measured at 1 min, 2 min, 5 min, 10 min, 15 min, 30 min, 1 h, 2 h, and 4 h. The reaction volumes were 300 μL and the systems consisted of 10 mM dCyd, 15 mM GTP, 15 mM MgCl$_2$, 50 mM potassium phosphate buffer (pH 7.5), and 0.69 nmol/mL (for dNK) or 9.45 nmol/mL (for dCK) of kinase. The reactions were performed at 37 °C and 500 rpm. Samples of 20 μL were quenched with 60 μL of acetonitrile, centrifuged at 16,000× *g* at 4 °C for 10 min and analyzed with HPLC-MS.

### 4.7.2. Optimal Temperature

The effect of temperature on dCMP synthesis was evaluated at 4 °C, 20 °C, 30 °C, 37 °C, 45 °C, 50 °C, 60 °C, 70 °C, and 80 °C. The reaction volume was 50 μL and the system consisted of 10 mM dCyd, 15 mM GTP, 15 mM MgCl$_2$, 50 mM potassium phosphate buffer (pH 7.5), and 0.69 nmol/mL dNK or 9.45 nmol/mL dCK. Syntheses were performed at 500 rpm for a duration of 5 min. Samples were quenched with 150 μL of acetonitrile, centrifuged at 16,000× *g* at 4 °C for 10 min, and analyzed with HPLC-MS.

### 4.7.3. Optimal pH

To evaluate the influence of the reaction medium's pH on dCMP synthesis, 50 mM potassium phosphate buffer with pH values of 5.5, 6.0, 6.5 and 7.0, and 50 mM tris-HCl buffer with pH values of 7.0, 7.5, 8.0, 8.5 and 9.0 were utilized. The reaction solution of 50 μL contained 10 mM dCyd, 15 mM GTP, 15 mM MgCl$_2$, 50 mM buffer, and 0.69 nmol/mL dNK or 9.45 nmol/mL dCK. The reactions took 5 min at 37 °C with a mixing speed of 500 rpm. Syntheses were terminated by adding 150 μL of acetonitrile, mixtures were centrifuged at 16,000× *g* at 4 °C for 10 min, and analyzed with HPLC-MS.

### 4.7.4. Optimal Nucleoside Concentration

The effect of 2′-deoxycytidine concentration on dCMP yields was assessed by mixing reaction solutions that were composed of 1 mM, 2.5 mM, 5 mM, 10 mM, 20 mM, 30 mM, or 50 mM dCyd, 30 mM GTP, 15 mM MgCl$_2$, 50 mM potassium phosphate buffer (pH 7.5) and 0.69 nmol/mL dNK or 9.45 nmol/mL dCK, with a total volume of 50 μL. After 5 min of mixing at 37 °C and 500 rpm, the reactions were halted by adding 150 μL of acetonitrile. The mixtures were centrifuged at 16,000× *g* at 4 °C for 10 min and analyzed with HPLC-MS.

### 4.7.5. Optimal Phosphate Donor Concentration

To determine how GTP amount in the reaction solution influences 2′-deoxycytidine conversion into 2′-deoxycytidine 5′-monophosphate, 8 different GTP:dCyd molar ratios were chosen: 1:4, 1:2, 1:1, 3:2, 2:1, 5:2, 3:1, and 5:1. The reaction mixtures made up a total volume of 50 μL and consisted of 10 mM dCyd, 2.5 mM, 5 mM, 10 mM, 15 mM, 20 mM, 25 mM, 30 mM, and 50 mM GTP, 15 mM MgCl$_2$, 50 mM potassium phosphate buffer (pH 7.5), and 0.69 nmol/mL dNK or 9.45 nmol/mL dCK. Reactions were executed at 37 °C and 500 rpm. After 5 min, 150 μL of acetonitrile were added, the samples were centrifuged at 16,000× *g* at 4 °C for 10 min and analyzed with HPLC-MS.

### 4.7.6. Optimal Nucleoside Kinase Concentration

To evaluate the effectiveness of the recombinant enzymes, five different enzyme concentrations were utilized for the phosphorylation of dCyd. The reaction mixtures contained 10 mM dCyd, 15 mM GTP, 15 mM MgCl$_2$, 50 mM potassium phosphate buffer (pH 7.5), and 0.07 nmol/mL, 0.14 nmol/mL, 0.34 nmol/mL, 0.69 nmol/mL, 1.37 nmol/mL, 2.06 nmol/mL (for dNK) or 1.89 nmol/mL, 4.72 nmol/mL, 9.45 nmol/mL, 18.9 nmol/mL, 56.7 nmol/mL, 94.5 nmol/mL, 189 nmol/mL (for dCK) kinase. Phosphorylation reactions were performed at 37 °C and 500 rpm for 5 min. Syntheses were quenched by adding 150 μL of acetonitrile, and the insoluble material was removed by centrifugation (16,000× *g*) at 4 °C for 10 min and analyzed with HPLC-MS.

### 4.8. Substrate Specificities of the Nucleoside Kinases

Substrate specificity assays of the nucleoside kinases were performed in 50 μL reaction mixtures that contained 10 mM nucleoside, 15 mM GTP, 15 mM MgCl$_2$, 50 mM potassium phosphate buffer (pH 7.5), and 0.69 nmol/mL dNK or 9.45 nmol/mL dCK. Reactions were performed in a thermomixer at 500 rpm with a temperature of 37 °C for 24 h. Then, the reaction solutions were diluted 4 times with acetonitrile, centrifuged at 16,000× *g* at 4 °C for 10 min and analyzed using TLC and HPLC-MS. Kinase specificities were determined based on the generation of corresponding nucleoside 5′-monophosphates. Reaction yields were calculated by utilizing HPLC-MS chromatogram areas. The nucleosides used for the experiments are listed in Tables 1 and 2.

### 4.9. Larger-Scale Syntheses of Nucleoside 5′-Monophosphates

Reaction conditions for the larger-scale syntheses of nucleoside 5′-monophosphates were based on previously described optimization assays. The progress of nucleoside phosphorylation was observed using TLC analysis. The initial reaction mixtures contained 50 mM nucleoside, 5 mM GTP, 15 mM MgCl$_2$, 100 mM acetyl phosphate, 15 mL 50 mM potassium phosphate buffer (pH 7.5), 0.46 nmol/mL ACK, and 9.45 nmol/mL dCK or 0.34 nmol/mL dNK. Reaction mixtures were incubated in round-bottom flasks at 37 °C and mixed by hand every few hours. An additional 1.5 mL of 1 M acetyl phosphate, 15 μL of 20 mg/mL ACK, and 150 μL of 12 mg/mL dCK or 1 mg/mL dNK were added after 24 h and 48 h. After executing the reactions for a total of 72 h, the mixtures were quenched with 15 mL of acetonitrile and centrifuged at 16,000× *g* at 4 °C for 10 min. Soluble fractions were collected and dried in a rotary evaporator. The remaining dry material was dissolved in 5 mL of water and loaded onto a chromatography column that contained 30 mL of DEAE Sephadex A-25 equilibrated with water. The ion-exchange chromatography was started with 150 mL of water, and after the unphosphorylated nucleoside was completely washed out, 200 mL of 0.05 M NaCl was used to elute the nucleoside monophosphate. Fractions containing the desired product were pooled and dried using a rotary evaporator. To further purify the collected monophosphate, the dry material was dissolved in 5 mL of water and loaded onto a Grace C-18 (12 g) reverse-phase column equilibrated with water. The nucleoside monophosphate was eluted using water. The fractions containing the product were pooled and dried using a rotary evaporator. The purification process was monitored using TLC analysis. The yields of purified nucleoside 5′-monophosphates were calculated using UV spectra of corresponding nucleosides. Compound structures were confirmed by HPLC-MS and NMR analyses. The nucleosides used for phosphorylation reactions were 2′-deoxycytidine and N$^4$-acetyl-2′-deoxycytidine for dCK and 2-thiouridine for dNK.

**Supplementary Materials:** The following supporting information can be downloaded at: https://www.mdpi.com/article/10.3390/catal12111401/s1, Figure S1: SDS-PAGE of the purified enzymes; Figure S2: (a) Catalytic verification of dCK and dNK using TLC analysis; (b) HPLC-MS analyses of 2′-deoxycytidine phosphorylation by dNK. References [23,37,38] are cited in the Supplementary Materials.

**Author Contributions:** Conceptualization, M.K. and R.M.; methodology, M.K.; validation, M.K. and K.B.; formal analysis, M.K., K.B., D.Š., N.U. and R.M.; investigation, M.K, K.B., D.Š. and N.U.; resources, R.M.; writing—original draft preparation, M.K.; writing—review and editing, M.K. and R.M.; visualization, M.K.; project administration, R.M.; funding acquisition, R.M. All authors have read and agreed to the published version of the manuscript.

**Funding:** This research was funded by the European Structural and Investment Funds via Central Project Management Agency of Lithuania, grant number 01.2.2-CPVA-K-703-03-0023.

**Data Availability Statement:** Not applicable.

**Conflicts of Interest:** The authors declare no conflict of interest.

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
