# Peer review of "Enzymatic Synthesis of Modified Nucleoside 5′-Monophosphates"

_catalysts, doi:10.3390/catal12111401_

Round 1
Reviewer 1 Report
The paper by Kaplounaite et al. reports on the comparison of two nucleoside kinases, D. melanogaster deoxynucleoside kinase and B. subtilis deoxycytidine kinase, respectively. These two biocatalysts have been employed in the phosphorylation of a panel of natural and modified nucleosides. Finally, three nucleosides were synthesized on larger scale employing a GTP regeneration system.
In my opinion, the results presented in this paper do not provide a significant contribution to the enzyme catalysis field. In the small scale reactions conversions 50-99% are claimed while in the larger scale reactions the final yield reached is around 5%, after different additions of biocatalysts. I think that some optimization/reaction engineering is needed in order to propose a preparative scale.
I would be available to reconsider this paper after major revisions.
Major revisions
1) Introduction:
- the Authors should add more appropriate references in the introduction in order to better define the state of the art and the background of biocatalyzed-nucleosides phosphorylation. Wild type nucleoside kinases, as the ones reported in this paper, have been used more recently (2017-2020) (e.g. Drosophila melanogaster, Dictyostelium discoideum deoxyadenosine kinase, human deoxycytidine kinase) also in preparative scale phosphorylations with better isolated yields than the ones reported in this paper.
2) Results:
- paragraph 2.1
- I cannot understand why the authors didn’t performed a standard activity assay after protein purification in order to determine the volumetric activity (IU/mL) or the specific activity (IU/mg) of each enzyme preparation. They simply performed the phosphorylation of 2ʹ-deoxycytidine and state that the enzyme are active. In my opinion in order to further perform a correct comparison between the two enzymes the same amount of enzymatic units should have been added in the reaction mixture. The two proteins are added in completely different amounts: 0.69 nmol/mL for dNK and 9.45 nmol/mL for dCK.
-In Figure S2 the retention time is reported in seconds but it should be in minutes at least looking to the x axe of the chromatogram.
- paragraph 2.2
- I would suggest to add also in the text and not just in the methods section the parameters that are maintained constant during the different studies.
Example: Impact of pH of reaction medium for dNK and dCK activity was evaluated using potassium phosphate (pH 5.5–7.0) and tris-HCl (pH 7.0–9.0) buffers. Reaction mixtures containing dNK reached the highest deoxycytidine monophosphate yield (50%) in the 5 min timeframe when reaction medium had a pH value of 8.0, although similar yields can be obtained in the pH range of 7.5–8.5. It is evident from figure 2c that dNK activity dropped drastically when pH reached 9.0. Kinase dCK possesses similar properties as dNK. The enzyme reached the greatest efficiency at pH 8.5, although similar deoxycytidine monophosphate yields, reaching 50%, can be achieved using reaction medium pH that ranges from 8.0 to 9.0.
Which temperature was used in the pH study? Which concentration of substrate? Amount of enzymes? etc. It is quite difficult to follow the results if one has to go every time to methods section to see all this details.
- Figure 2: I would suggest to add more numbers on the x axes especially in graphic a and b. The English in the legend of the figure should be revised.
- at the end of the paragraph 2.2 the authors should introduce a final sentence showing the best reaction conditions based on the optimization study.
- paragraph 2.3
- why the standard reaction with 2ʹ-deoxycytidine was studied in a range of 5 min to 120 min with maximum conversion after 30 min and in the screening 24 hours were chosen as end-point? Did the authors analyzed also intermediate points? Moreover in the methods section the authors state that the reactions were stirred manually, so did they stirred the reactions manually also during the night?
- paragraph 2.5
- the authors should explain why they have to add 100 mM of acetylphosphate to the reaction mixtures every 24 h to “better regenerate GTP”. Is this molecule not stable in the reaction conditions?
- I suppose that the acetic acid that is formed during the reaction is lowering the pH of the reaction mixture, are the proteins stable and or active once the reaction is going on? Have the authors tried to measure the pH of the reaction at the end-point (72 h) or before the additions of fresh enzymes? I suppose that the low pH generated during the reaction, and the inactivation of enzymes, is the cause of low isolated yields in the preparative synthesis even after different additions of fresh enzymes. Can the authors comment on the low isolated yields when the conversions for the selected nucleosides were around 90%.
4) Materials and Methods
-paragraph 4.5
-The authors can specify which percentage of acrylamide is the SDS-gel they have used. The same information should be added into the legend of the SDS gel present in SI
-paragraph 4.6
- Generally a standard activity assay is used in order to determine and quantify the enzymatic activity of the preparation. The authors should quantify the activity of their enzymes and not just state that the enzymes are active after their preparation.
paragraph 4.7.1
- There is a typo error in line 454 “The reaction was volumes were 300 μL”
paragraph 4.9
- Why the authors didn’t used a magnetic stirrer to mix the reactions?
Reviewer 2 Report
26-Oct-2022
Manuscript ID: catalysts-2005004
Type of manuscript: Article
Title: Enzymatic synthesis of modified nucleoside 5ʹ-monophosphates
Authors: Martyna Koplūnaitė *, Kamilė Butkutė, Dominykas Špelveris, Nina Urbelienė, Rolandas Meškys
Submitted to section: Biocatalysis,
Comments to the author
In this paper, the basic properties and substrate specificities of two kinases of different origin, deoxynucleoside kinase and deoxycytidine kinase, are reported. These two enzymes phosphorylate hydroxyl groups at the 5' position, and they report that the reaction proceeds with both purine and pyrimidine base sites of nucleosides. However, they report that the introduction of a bulky functional group significantly reduces the reaction activity and makes it difficult for the enzyme to enter the active site. The effect of the hydroxyl group of ribose for the reaction is also discussed.
The quantity and quality of the experiments are sufficient, and although good yields have not been obtained, the authors have attempted to synthesize the compounds on a large scale. In view of the above, this paper is judged to be sufficient for publication in Catalysis. We would like to point out a few corrections.
1. Although much work has been done on substrate specificity, there seems to be little consideration of the structure around the active site of the enzyme and the shape of the pocket.
2. From the experiments on substrates 43 and 44, we could see the importance of the 5' hydroxyl group. The experiment on substrate 57 also showed the same result. However, I have one question: if we put a primary hydroxyl group at the 2' and 3' positions, do these compounds become a substrate with dCK and dNK? I could not determine from these data whether this enzyme recognized the ribose ring and selected only the 5' position, or whether it was looking for a primary hydroxyl group.
3. In the Supplementary Data section of the Nucleoside Synthesis, there were many cases where the N in NMR was replaced by B. So please correct these spellings.
Round 2
Reviewer 1 Report
I do not have further comments.